# Psychological Distress, Dental Health, and Dental Fear among Finnish University Students: A National Survey

**DOI:** 10.3390/ijerph181910245

**Published:** 2021-09-29

**Authors:** Vesa Pohjola, Kristina Kunttu, Jorma I. Virtanen

**Affiliations:** 1Medical Research Centre, Oulu University Hospital, 90220 Oulu, Finland; vesa.pohjola@oulu.fi; 2Unit of Oral Health Sciences, University of Oulu, 90014 Oulu, Finland; 3Finnish Student Health Service, 00260 Helsinki, Finland; kristina.kunttu@historia-memoria.fi; 4Institute of Dentistry, University of Turku, 20014 Turku, Finland; 5Department of Clinical Dentistry, University of Bergen, 5020 Bergen, Norway

**Keywords:** dental fear, oral health, psychological distress, students

## Abstract

The aim of this study was to investigate the association between dental fear, psychological distress, and perceived symptoms of teeth controlled for age, gender, educational sector, and tobacco use. The data from the Finnish University Student Health Survey 2016 targeting students (*n* = 10,000) of academic universities and universities of applied sciences were used. Psychological distress was measured with the Clinical Outcomes in Routine Evaluation 10 (CORE-10) and the General Health Questionnaire 12 (GHQ-12) and dental fear with the question ‘Do you feel scared about receiving dental care?’. The study included 3110 students. In logistic regression analyses those with psychological distress (measured with CORE-10 and GHQ-12) and those reporting teeth-related symptoms were more likely than their counterparts to have high dental fear. In gender-specific analyses men with psychological distress (measured with CORE-10) and women with teeth-related symptoms were more likely to have high levels of dental fear. Finnish university students with psychological distress and teeth-related symptoms were more likely to experience higher levels of dental fear than their counterparts were. The results of this study support possible common vulnerability factors that dental fear and other psychological disorders may share.

## 1. Introduction

The traditional explanation for dental fear is based on learning theory [1]. People have learned dental fear after suffering painful experiences, having witnessed traumatic events or having heard frightening stories about dental care. However, many people with no dental fear have had negative dental experiences, and some people with considerable fear fail to recall any traumatic incidents [2]. Beside the exogenous components, there are also endogenous components in dental fear [3]. Cognitive models of vulnerability provide ways of understanding the differences in the expression of dental fear despite similar experiences of dental care [4,5].

The symptoms of depression and anxiety can collectively be termed psychological distress, which can be measured e.g., with the Clinical Outcomes in the Routine Evaluation 10, CORE-10 [6,7,8] and the General Health Questionnaire 12, GHQ-12 [9,10,11]. Positive associations between psychological distress and dental fear [12] and depressive and anxiety disorders and dental fear [13,14,15,16,17,18,19,20,21] have been found. Gender differences in the prevalence of depressive and anxiety disorders and dental fear have also been reported [12,16,17,18,19]. However, most studies have been conducted either on clinical samples of dental patients [12,17,22] or among dental patients with high dental anxiety [21,23,24] and more population level studies are needed. According to our knowledge, this is the first study exploring the association between dental fear and psychological distress measured with CORE-10 and GHQ-12 using data from comprehensive student population study.

People with high dental fear have more frequently oral health problems than those with lower level of dental fear [25,26,27]. The most obvious consequence of dental fear is irregular visiting of a dentist and symptom based dental visits for a problem or for the relief of pain [23,26,27,28]. The connection between dental fear and irregular visiting of a dentist and problems of oral health has been explained by a vicious cycle of dental fear [26,29]. Additionally, pain has been connected to psychological distress. Depressed patients have reported more pain before entering the dental operatory than nondepressed patients [30]. Tobacco use is also associated with psychological distress; anxiety and depression are more commonly reported by smokers than non-smokers [31,32]. Smokers have also reported dental fear more frequently than non-smokers [28,33]. Furthermore, other unfavorable oral health habits (e.g., irregular tooth brushing) have been connected with psychological distress, indicating a greater risk for oral health problems among those with psychological distress [34].

When starting university studies, students enter a new life situation where they have more responsibility for their everyday lives. This new life situation may also have effects on psychological distress and have impacts on health later in life [35]. The aim of this study was to investigate the association between dental fear and psychological distress (measured with CORE-10 and GHQ-12) and perceived symptoms of the teeth controlling for age, gender, educational sector, and tobacco use. Our hypotheses were that students with high psychological distress and teeth-related symptoms would be more likely to report high dental fear compared to students with no or low psychological distress and no teeth-related symptoms.

## 2. Materials and Methods

### 2.1. Study Design and Participates

The Finnish Student Health Service (FSHS) conducts a national University Student Health Survey (USHS) every four years. Participation in the USHS is voluntary. The USHS aims to investigate students’ physical, mental and social health, health-related behavior, and the use of health services. Study permission was granted from the FSHS authorities and the Ethics Committee of the University of Turku (reference 35/2015 26.10.20215) approved the study. We used the cross-sectional USHS 2016 data in this study [36].

The study population was Finnish undergraduate students under 35 years of age in the Finnish academic universities (Univ) and universities of applied sciences (UAS). In all, 14 out of 15 Univ and 24 out of 26 UAS in Finland participated. The USHS report presents the inclusion and exclusion criteria as well as the handling of missing data for the study [36]. The sample included 10,000 students (UAS 5004; Univ 4996). Altogether, 47% of the sample were male. The respondents received an initial invitation and five reminders by e-mail. The third reminder also served as a repeat survey and was sent as a posted questionnaire in paper form. The questionnaire was in Finnish and Swedish, which are the two official languages in Finland. The overall response rate was 31% (men 22% and women 39%). When compared to the national statistics of university students [37], the respondents represented well the target population (e.g., for age, educational sector, Univ, UAS, Univ faculty and UAS degree program), except for the underrepresentation of men. To correct the underrepresentation of men USHS professionals applied statistical measures to count weights for men and women separately for the educational sectors [36]. After weighting adjustments, the sample represented well the target population.

### 2.2. Dependent Variable

Dental fear was queried by the question: “Do you feel scared about receiving dental care?”. The reply alternatives were “Not at all”, “Somewhat” and “Very”. The alternatives ‘Not at all’ and ‘Somewhat’ were then combined into a category indicating ‘Low or no’ fear and ‘Very’ was used as the category for high level of fear. As the question used in this study directly refers to ‘dental care and dental fear’, it can be used as a measure of dental fear [38]. The validity and reliability of assessing dental fear with a single question has been approved in previous studies in the Nordic countries and Japan [39,40,41]. Appendix A shows details for the dependent variable and the independent variables.

### 2.3. Independent Variables

Psychological distress was assessed with the Clinical Outcomes in Routine Evaluation 10 (CORE-10), which has been validated and shown to have good psychometric properties [6,7,8]. CORE-10 included the following 10 items: (1) Feeling tense, anxious or nervous, (2) Having someone to turn to for support, (3) Able to cope when things go wrong, (4) Talking to people has felt to be too much, (5) Feeling panic or terror, (6) Planning to end my life, (7) Difficulty sleeping, (8) Feeling despair or hopeless, (9) Feeling unhappy, and (10) Having distressing unwanted images or memories. These questions were rated on a 5-point scale ranging from 0 (‘not at all’) to 4 (‘most or all the time’). The items 2 and 3 were reverse scored on the 5-point scale. The possible total scores may range from 0 to 40. We used the cut-off score ≥ 11, as scores of 11 or above have been considered as indicative of general or clinically significant psychological distress [6]; in the Finnish population, a score ≥ 11 has been indicative of daily or nearly daily anxiety [8]. Based on this cut-off score, we formed two categories (scores ≤ 10 representing a lower or negative CORE-10 score and scores ≥ 11 representing higher or positive CORE-10 score).

The General Health Questionnaire 12 (GHQ-12) was also used to assess psychological distress. The 12-item version GHQ-12 is a widely used brief screening instrument to assess mental well-being and psychological symptoms [10]. The reliability, construct and content validity of the GHQ have been regarded as good [9,10,11]. The GHQ-12 had 12 questions: (1) Able to concentrate, (2) Loss of sleep over worry, (3) Playing a useful part, (4) Capable of making decisions, (5) Feel constantly under strain, (6) Could not overcome difficulties, (7) Able to enjoy day-to-day activities, (8) Able to face problems, (9) Feeling unhappy and depressed, (10) Losing confidence, (11) Thinking of self as worthless, and (12) Feeling reasonably happy. These 12 questions were rated on a four-point scale (see Appendix A) and the bi-modal (0-0-1-1) scoring method giving a total score of 12 was used. As scores 4 or above have been considered to indicate the presence of a mental disorder [10], we used the cut-off score ≥ 4 and formed two categories (scores ≤ 3 representing a lower or negative GHQ-12 score and scores ≥ 4 representing a higher or positive GHQ-12 score). 

Dental health and teeth-related symptoms were assessed with the question “Have you had tooth problems (shooting pain, toothache) over the past month (30 days)?” with answer options were being “Not at all”, “Every now and then”, “Weekly”, and “Daily or almost daily”. For the analyses, two categories were used: “Not at all” and “Sometimes, weekly or daily” (answer options “Every now and then”, “Weekly”, and “Daily or almost daily”).

Smoking was asked using the question “Do you smoke?” and snuff use with question “Do you use snuff?”. Answer options to both questions were “Not at all”, “Yes previously but I have quit”, “Less than once a week”, “Weekly but not daily”, and “Daily”. Later, two categories were used: “Not at all or having quit” (including answer options “Not at all” and “Yes previously but I have quit”), and “Yes” (answer options “Less than once a week”, “Weekly but not daily”, and “Daily”). Subsequently, the questions of smoking and snuff use were combined into “Tobacco use” with two answer categories: “Not at all or having quit” and “Yes”; these categories included the same answer options as the questions on smoking and snuff use.

The students belonged to the educational sectors “Universities” or “Universities of applied sciences”. Age was categorized age into three groups: 19–24, 25–30, and 31–35 years. This was done because age groups are different, e.g., the youngest age group has entered the Univ or UAS directly or shortly after graduating from high school and the other age-groups may have been, e.g., working or studying something else before the university studies.

### 2.4. Statistical Analysis

Cross-tabulations with Chi-squared tests were used to analyze associations between background variables, psychological distress (measured with CORE-10 and GHQ-12), tobacco use, teeth-related symptoms, and dental fear. After checking for multicollinearity, logistic regression analyses were conducted with dental fear as the dependent variable, and age, gender, educational sector, tobacco use, symptoms of teeth, and psychological distress (measured with CORE-10) as covariates. As gender strongly impacted dental fear and psychological distress, gender-specific logistic regression analyses were conducted in the same way as described above. After the logistic regression, analyses were repeated using GHQ-12 instead of CORE-10 as a measure of psychological distress. The results are presented with adjusted odds ratios (OR) and their 95% confidence intervals (95% CI). IBM SPSS Statistics for Windows, Version 22.0. Armonk, NY, USA: IBM Corp. was used for all statistical analyses and *p*-values < 0.05 were considered statistically significant.

## 3. Results

Women reported high dental fear approximately three times more often than men (Table 1). Women also reported psychological distress more frequently than men. One out of four men and one out of three women scored positive in CORE-10. A positive score in the GHQ-12 was found among more than one in five men and among one in four women. Teeth-related symptoms were reported by one out of four among both genders.

Students reporting psychological distress (scored positive in CORE-10 or GHQ-12) reported high dental fear more commonly than those not reporting psychological distress (Table 2). Additionally, students with teeth-related symptoms or tobacco use had high dental fear more often than those not having teeth-related symptoms or not using tobacco.

Students with psychological distress had teeth-related symptoms more often than those not reporting psychological distress (Table 3). Those with psychological distress were also using tobacco more commonly.

According to logistic regression analyses (Table 4) when controlling for gender, educational sector, and tobacco use, those with psychological distress (measured with CORE-10) and reporting teeth-related symptoms were more likely to have high dental fear (OR = 1.5; CI = 1.1–2.0 and OR = 2.2; CI = 1.7–3.0, respectively) than those not having psychological distress and not reporting teeth-related symptoms. In the gender-specific logistic regression analyses among men, those with psychological distress (measured with CORE-10) were more likely to have high dental fear (OR = 2.2; CI = 1.2–3.8). Among women, those reporting teeth-related symptoms were more likely to have high dental fear (OR = 2.2; CI = 1.6–3.1).

In the logistic regression analyses, when replacing the CORE-10 by GHQ-12 (Table 5), the results for the whole study population were similar as described above, but in the gender-specific logistic regression analyses psychological distress (measured with GHQ-12) was not significantly associated with dental fear among men or women.

## 4. Discussion

The students with psychological distress and reporting teeth-related symptoms were more likely to have high dental fear than the students without psychological distress or teeth-related symptoms, when controlling for age, gender, educational sector, and tobacco use. However, there were gender differences in these associations. Among men, the students with psychological distress (measured with CORE-10) were more likely to have high dental fear, but teeth-related symptoms were not significantly associated with dental fear. Among women, the students reporting teeth-related symptoms were more likely to have high dental fear, but psychological distress was not significantly associated with dental fear.

Our finding that the students with psychological distress were more likely to have high dental fear compared to the students without psychological distress is in concordance with previous studies, e.g., recently a positive association was found between dental anxiety and psychological distress measured with the Brief Symptom Inventory-18 [12]. Our findings are also in line with the results of the studies where high dental fear has been positively connected with depressive and anxiety disorders [13,14,15,16,17,18,19,20,21]. However, it was surprising that in the gender-specific analyses, psychological distress (measured with CORE-10) was associated with dental fear among men, but not among women. The prevalence of high dental fear is higher among women than men [12,16,17,18,19]. It is possible that among women the high dental fear group was more heterogeneous and included several sub-groups in the etiology of dental fear. Thus, in our study, the origin of dental fear in the high dental fear group could have been partly different among men and women, and this could have affected the association between dental fear and psychological distress. It was also surprising that among men, when psychological distress was measured with CORE-10 it was associated with dental fear, while no significant association was found with GHQ-12. It is likely that CORE-10 and GHQ-12 measure slightly different aspects of psychological distress and the lower number of men with psychological distress measured with GHQ-12 than CORE-10 may have affected the associations between dental fear and psychological distress measured with these two measures of psychological distress.

Women reporting teeth-related symptoms were more likely to have high dental fear than women without symptoms. This finding is in concordance with previous results indicating that people with high dental fear have more often oral health problems than those with lower level of dental fear [25,26,27]. It was interesting that the association between teeth-related symptoms and dental fear was not seen among men. As men visit a dentist less regularly than women [42], they are likely to have teeth-related symptoms even without the ‘vicious cycle’ of dental fear causing the deterioration of oral health [26,29]. This could have diminished the association between dental fear and teeth-related symptoms among men. Additionally, the ‘vicious cycle’ of dental fear includes feelings of shame and embarrassment, which may play a role in psychological distress, if the shame and embarrassment results from avoidance of the dentist and consequent tooth decay [43]. This could have strengthened the associations between teeth-related symptoms, psychological distress and dental fear among women. Additionally, in a recent study unfavorable health and oral health habits (like smoking, irregular meals, and irregular tooth brushing) were related to psychological distress. This can point to a greater risk for oral health problems among those with psychological distress [34].

Tobacco use was associated with dental fear, the students using tobacco were more likely to have high dental fear than those not using. Smoking and nicotine dependence have been associated with anxiety disorders in general [32]. The association between tobacco use, nicotine dependence and anxiety disorders could be explained by three relationships; smoking and nicotine dependence leads to increased anxiety disorders, the reverse association, or a shared vulnerability model. There could be vulnerability factors increasing smoking, nicotine dependence and anxiety disorders [32].

Common vulnerability factors shared by psychological disorders may explain the connection between dental fear and psychological distress. Explanation of the factors that contribute to the onset and maintenance of psychological problems can be provided by cognitive vulnerability models [4]. In a longitudinal study it was found that women who reported lower levels of dental fear were less distressed [44]. Among part of the people with psychological distress, the cognitive vulnerability may partly explain dental fear. However, as most of the people reporting psychological distress did not have dental fear, other factors such as conditioning experiences might play a more important role than psychopathology in the acquisition of dental fear. A proportion of people may have a constitutional vulnerability to developing anxiety disorders (e.g., dental fear) and psychological distress [4,14]. These people might benefit from the co-operation of dentists and psychologists in the treatment of dental fear.

Practically all Finnish academic universities and universities of applied sciences participated the study accounting for the remarkably large sample. The students’ participation was enhanced by sending the questionnaires as emails (followed by five reminders). The participants were able to skip some questions more easily on web than in a face-to-face interview. Yet, more missing values have been discovered in paper-based than in web-based surveys and both forms of surveys have similar levels of selection bias [45,46]. Although the paper-based questionnaires may have slightly higher response rates, the web-based questionnaires are less expensive to administer, which makes them ideal for large-scale studies. The response rate of this study was similar to other web-based surveys [45,46]. Generally, women take part in studies more frequently than men do. This was seen in our study too; to balance this weighting adjustments were applied [36]. When compared with the national statistics [37], the participants of this study represented the target population for age, study field, faculty, and educational sector. Furthermore, no remarkable downward or upward trend was noticed when comparing the results of health and health habits of USHS 2016 with those of previous USHS studies [36]. This suggests that the composition of the participants has not changed considerably.

When compared with multi-item questionnaires, single questions of dental fear have been found valid and reliable [39,40,41]. Single questions of dental fear are easier and faster to answer in comprehensive and large studies. People with dental fear are characteristically afraid of dental treatment [47] and often avoid dental care [23,26,27,28]. Thus, collecting information about dental fear out of dental environment reduces the risk of selection bias. The question of dental fear we used in this study has given similar prevalence of dental fear when compared to the single questions referring to visiting a dentist [19,27,33].

In our study, CORE-10 and GHQ-12 were used as measures of psychological distress. Both CORE-10 and GHQ-12 have been validated in Finnish [8,10]. The findings of this study cannot though be generalized to all young Finnish adults since the survey focused on university students. In addition, it is important to keep in mind that one cannot draw causal conclusions of cross-sectional studies.

## 5. Conclusions

High dental fear was more likely experienced by students with psychological distress or teeth-related symptoms than among students without psychological distress or students free of teeth-related symptoms. Dental fear and other psychological disorders may share common vulnerability factors. Questions about oral health and dental fear ought to be included in student health questionnaires. When healthcare professionals (e.g., health nurses, physicians and psychologists) meet students with psychological distress, oral health issues should also be discussed.

## Figures and Tables

**Table 1 ijerph-18-10245-t001:** Age groups, educational sector, dental fear, GHQ-12, CORE-10, perceived teeth-related symptoms and tobacco use of the study population by gender (*n* = 3090).

	Total % (*n*)	Men % (*n*)	Women % (*n*)	*p*-Value ^¤^
Gender (*n* = 3079)	100 (3079)	47.1 (1451)	52.9 (1628)	
Age (*n* = 2877)				0.001
19–24	50.3 (1447)	47.2 (647)	53.1 (800)	
25–30	40.8 (1174)	44.3 (608)	37.6 (566)	
31–35	8.9 (256)	8.5 (116)	9.3 (140)	
Educational sector (*n* = 3080)				0.452
Univ	59.4 (1829)	58.7 (852)	60.0 (977)	
UAS	40.6 (1251)	41.3 (600)	40.0 (651)	
Tobacco use (*n* = 3002)				<0.001
Not at all or quit	77.7 (2333)	73.3 (1024)	81.6 (1309)	
Yes	22.3 (669)	26.7 (373)	18.4 (296)	
Teeth-related symptoms (*n* = 3079)				0.585
Not at all	73.6 (2267)	74.1 (1075)	69.8 (1192)	
Sometimes/weekly/daily	26.4 (812)	25.9 (376)	26.8 (436)	
Dental fear (*n* = 3015)				<0.001
No or low	92.2 (2781)	96.2 (1353)	88.8 (1428)	
High	7.8 (234)	3.8 (54)	11.2 (180)	
GHQ-12 (*n* = 3073)				0.001
Negative (≤3)	75.3 (2315)	78.1 (1131)	72.9 (1184)	
Positive (≥4)	24.7 (758)	21.9 (318)	27.1 (440)	
CORE-10 (*n* = 3071)				0.034
Negative (≤10)	72.5 (2228)	74.4 (1076)	70.9 (1152)	
Positive (≥11)	27.5 (843)	25.6 (371)	29.1 (472)	

GHQ-12: General Health Questionnaire-12; CORE-10: Clinical Outcomes in Routine Evaluation-10; Univ: academic universities; UAS: universities of applied sciences; ¤ Chi-square test.

**Table 2 ijerph-18-10245-t002:** Age, educational sector, GHQ-12, CORE-10, perceived teeth-related symptoms, and tobacco use by dental fear (*n* = 3013) among the Finnish university students.

	Dental Fear		
	No or Low % (*n*)	High % (*n*)	*p*-Value ^¤^
Age (*n* = 2837)			0.096
19–24	92.1 (1311)	7.9 (112)	
25–30	92.1 (1068)	7.9 (92)	
31–35	88.2 (224)	11.8 (30)	
Educational sector (*n* = 3014)			0.003
Univ	93.4 (1688)	6.6 (119)	
UAS	90.5 (1092)	9.5 (115)	
Tobacco use (*n* = 2988)			<0.001
Not at all or quit	93.3 (2169)	6.7 (155)	
Yes	88.4 (587)	11.6 (77)	
Teeth-related symptoms (*n* = 3015)			<0.001
Not at all (*n* = 2210)	94.1 (2080)	5.9 (130)	
Sometimes/weekly/daily (*n* = 805)	87.1 (701)	12.9 (104)	
GHQ-12 (*n* = 3013)			0.001
Negative (≤3)	93.2 (2108)	6.8 (155)	
Positive (≥4)	89.5 (671)	10.5 (79)	
CORE-10 (*n* = 3013)			<0.001
Negative (≤10)	93.4 (2039)	6.6 (143)	
Positive (≥11)	89.0 (740)	11.0 (91)	

GHQ-12: General Health Questionnaire-12; CORE-10: Clinical Outcomes in Routine Evaluation-10; Univ: academic universities; UAS: universities of applied sciences; ¤ Chi-square test.

**Table 3 ijerph-18-10245-t003:** Gender, age, educational sector, dental fear, teeth-related symptoms and tobacco use by CORE-10 and GHQ-12 (*n* = 3073) among the Finnish university students.

	CORE-10 Negative (≤10) % (n)	CORE-10 Positive (≥11) % (n)	*p* ¤	GHQ-12 Negative (≤3) % (n)	GHQ-12 Positive (≥4) % (n)	*p* ¤
Gender			0.034			0.001
Men	74.4 (1076)	25.6 (371)		78.1 (1131)	21.9 (318)	
Women	70.9 (1152)	29.1 (472)		72.9 (1184)	27.1 (440)	
Age			0.075			0.005
19–24	74.1 (1068)	25.9 (374)		77.4 (1116)	22.6 (326)	
25–30	70.5 (825)	29.5 (346)		72.0 (844)	28.0 (329)	
31–35	69.5 (178)	30.5 (78)		73.0 (187)	27.0 (69)	
Educational sector			0.449			0.744
Univ	72.1 (1315)	27.9 (510)		75.1 (1371)	24.9 (454)	
UAS	73.3 (914)	26.7 (333)		75.6 (944)	24.4 (304)	
Tobacco use			0.007			0.048
Not at all or quit	73.6 (1716)	26.4 (616)		76.0 (1772)	24.0 (561)	
Yes	68.3 (456)	31.7 (212)		72.2 (483)	27.8 (186)	
Teeth-related symptoms			<0.001			<0.001
Not at all	75.3 (1701)	24.7 (559)		77.0 (1743)	23.0 (520)	
Sometimes/weekly/daily	65.0 (527)	35.0 (284)		70.6 (572)	29.4 (238)	
Dental fear			<0.001			<0.001
No or low	73.4 (2039)	26.6 (740)		75.9 (2108)	24.1 (671)	
High	61.1 (143)	38.9 (91)		66.2 (155)	33.8 (79)	

CORE-10: Clinical Outcomes in Routine Evaluation-10; GHQ-12: General Health Questionnaire-12; Univ: academic universities; UAS: universities of applied sciences; *¤* Chi-square test.

**Table 4 ijerph-18-10245-t004:** Results of logistic regression analyses including CORE-10 as a measure of psychological distress, dental fear as a dependent variable (not at all or somewhat afraid = 0, very afraid = 1).

	OR	95% CI
All Participants		
Gender	3.5	2.5–4.8
Educational sector	1.4	1.1–1.9
Tobacco use	1.9	1.4–2.6
Teeth-related symptoms	2.2	1.7–3.0
CORE-10	1.5	1.1–2.0
Men (gender specific model)		
Age	0.4	0.2–0.7
Tobacco use	2.2	1.2–3.9
CORE-10	2.2	1.2–3.8
Women (gender specific model)		
Age	1.8	1.1–3.0
Educational sector	1.7	1.2–2.3
Tobacco use	2.0	1.4–2.8
Teeth-related symptoms	2.2	1.6–3.1

Variables entered: Age (1 = 22 years or older), Gender (1 = women), Educational sector (1 = UAS), Tobacco use (1 = sometimes, weekly or daily), Teeth-related symptoms (1 = sometimes, weekly or daily), CORE-10 (1 = 11 or more). All participants: Nagelkerke R Square 0.106; Men: Nagelkerke R Square 0.051; Women: Nagelkerke R Square 0.072.

**Table 5 ijerph-18-10245-t005:** Results of logistic regression analyses including GHQ-12 as a measure of psychological distress, dental fear as a dependent variable (not at all or somewhat afraid = 0, very afraid = 1).

	OR	95% CI
All participants		
Gender	3.5	2.5–4.8
Educational sector	1.4	1.1–1.9
Tobacco use	1.9	1.4–2.6
Teeth-related symptoms	2.3	1.7–3.0
GHQ-12	1.4	1.0–1.8
Men (gender specific model)		
Age	0.4	0.2–0.8
Tobacco use	2.1	1.2–3.7
Teeth-related symptoms	1.8	1.1–3.2
Women (gender specific model)		
Age	1.8	1.1–3.0
Educational sector	1.7	1.2–2.3
Tobacco use	2.0	1.4–2.8
Symptoms of teeth	2.3	1.6–3.1

Variables entered: Age (1 = 22 years or older), Gender (1 = women), Educational sector (1 = UAS), Tobacco use (1 = sometimes, weekly or daily), Teeth-related symptoms (1 = sometimes, weekly or daily), GHQ-12 (1 = 4 or more). All participants: Nagelkerke R Square 0.102; Men: Nagelkerke R Square 0.044; Women: Nagelkerke R Square 0.072.

## Data Availability

The data are available in the Finnish Social Science Data Archive where registered users can download data online according to the conditions set for this data. http://urn.fi/urn:nbn:fi:fsd:T-FSD3224 accessed on 27 September 2021.

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
