# Peer review of "Psychological Distress, Dental Health, and Dental Fear among Finnish University Students: A National Survey"

_ijerph, 2021, doi:10.3390/ijerph181910245_

Round 1

Reviewer 1 Report

Long sentences should be rephrased to allow a better understandable. The introduction and discussion should be extensively synthesized, removing repetitive concepts and reformulating too longer periods. The concepts expressed in the "independent variables" paragraph could be summarized and reported in an explanatory table.

Author Response

The manuscript has been checked and edited by a native English language expert.

In the Introduction long sentences have been parted into two.

We have rewritten parts of the discussion and deleted some sentences.

We have added an Appendix of the variables.

Reviewer 2 Report

Thank you for sharing this interesting manuscript with me. From my point of view, it is well-written, the data base is excellent and the manuscript has a clear focus. Therefore, I only have minor comments.

  • Table 1: Please include the total proportions in an additional column.
  • Please include in conclusions a perspective for practice. What do the findings mean? What are potential implications for universities? How could health professionals act?

Author Response

Total proportions column has been added to the Table 1.

We have added practical implications into the conclusion.

Reviewer 3 Report

I suggest avoiding the first person (we) as in We aimed to study (Abstract, line 12); According to our knowledge (Introduction, line 45), We aimed to study (Introduction, line 63), Our, hypotheses (introduction, line 66), We used (Materials and Methods, line 76), Our finding that those (Discussion, line 211), Our findings (Discussion, line 214), Thus, in our study (Discussion, line 224),

I suggest describing the variables using a table, it seems easier to understand.

Author Response

The manuscript has been checked by a native English language expert and he prefers to use active voice as recommended for modern medical research reports.

We rephrased ''We aimed'' with ''The aim of this study was'' in the Abstract and in the Introduction.

We have added an Appendix of the variables.

Reviewer 4 Report

This is a comprehensive study that included 3110 students at Finnish Universities about association between dental fear and psychological distress, teeth related symptoms and tobacco use. Dependent variable was dental fear and it was assessed with  a single question, and psychological distress was assessed using two instruments previously shown reliable and valid CORE 10 and GHQ12.

ABSTRACT

Consider removing the brief definition of psychological distress from the abstract.

Replace '' We aimed'' with ''The aim of this study was''

DISCUSSION

The beginning of the discussion could be better formulated. The first paragraph points out major findings of the study as it should, but consider not beginning with “those“. Generally those could be replaced with “the students“

Author Response

We have deleted definition of psychological distress from the Abstract.

We rephrased ''We aimed'' with ''The aim of this study was'' in the Abstract and in the Introduction.

We have used “the students” instead of “those”.
